# Injectable Hydrogel for Cu^2+^ Controlled Release and Potent Tumor Therapy

**DOI:** 10.3390/life11050391

**Published:** 2021-04-26

**Authors:** Chunyu Huang, Bei Chen, Mingzhu Chen, Wei Jiang, Wei Liu

**Affiliations:** 1Key Laboratory of Artificial Micro- and Nano-Structures of Ministry of Education, School of Physics and Technology, Wuhan University, Wuhan 430072, China; cyhuang@whu.edu.cn (C.H.); beichen@whu.edu.cn (B.C.); pearlc@whu.edu.cn (M.C.); 2Center for Precision Medicine, The Second Affiliated Hospital of Zhengzhou University, Academy of Medical Sciences, Zhengzhou University, Zhengzhou 450001, China

**Keywords:** disulfiram, copper ions, controlled release, injectable hydrogel, hierarchical microparticle and anti-tumor

## Abstract

Disulfiram (DSF) is an important drug for the treatment and management of alcohol dependency. This drug has been approved by US-FDA, and its activity against the tumor is dependent on copper ion (Cu^2+^). However, the copper toxicity (caused via external copper) and its intrinsic anfractuous distribution in the human body have adversely suppressed the mechanism of DSF in in vivo. In this study, we aimed to design an injectable hydrogel, as CRC (Cu^2+^ release controller) for the effective treatment of tumors. The hydrogels of agarose have been used for wrapping of CuCl_2_, and hierarchical microparticles (HMP) for the generation of CRC system. When the laser irradiations (808 nm) have been provided to the system, light energy is transferred into heat energy, which results in the hydrogel hydrolysis (reversible) due to the overheating effect. This is followed by a reaction with DSF (pre-injected) to suppress tumor progression. Hence, the CRC system brings innovative ideas for designing of a Cu^2+^ delivery system.

## 1. Introduction

An effective advancement has been made in the detection and cure of carcinoma, however, cancer is a lethal disease across the globe [1,2,3,4,5], and chemotherapy is still frequently used for the treatment of carcinoma in the clinical field [3,6,7,8,9], although it has adverse and harmful side effects, i.e., multidrug resistance [10,11,12,13,14,15]. Therefore, it is needed to find out new effective drugs against carcinoma with relatively fewer side effects. Chen at al. [16] designed a novel photothermal selenium-coated tellurium nano heterojunctions therapy to eradicate tumor. Xing and co-workers [17] studied a new black phosphorus (BP)/cellulose hydrogel for photothermal agents and obtained a good effect in tumor treatment. Zhang et al. [18,19] designed different novel two-dimensional materials for cancer theranostics. These results show good potential, although new drug development against carcinoma is extremely challenging because of high research cost and failure rates along with long testing periods [20,21,22]. On the other hand, the finding of new drugs against cancer among the approved drugs that have been used for the treatment of other diseases is safe and is an effective approach [23]. Recently, a large number of drugs, i.e., metformin, aspirin, and disulfiram (DSF), have been indicated to have potential against cancer [24].

DSF is an effective drug to cure alcohol dependence, and in recent times it has been revealed that DSF is a remarkable drug against carcinoma, particularly for breast carcinoma [24]. The underlined drug mechanism is dependent on copper. DSF is a dimer (disulfide), which consists of DTC (dithiocarbamate) moieties that are well known for significant metal-chelating activity [25]. DSF has been metabolized to DTC in the physiological environment, which chelates metal ions in a different range, mainly copper ions, which form CuETs (DTC−copper complexes) [23]. The underlined complex has been indicated to possess a remarkable potential against tumor development and progression in comparison with the original DSF. It has been recognized that the CuET complex interacts with the nuclear protein localization-4 (NPL4) in a compact manner and causes its aggregation. Therefore, the p97−NPL4−ubiquitin fusion degradation protein 1 (UFD1) cascade, which has a key contribution to cellular metabolism, is adversely interrupted and results in apoptosis [23,26,27].

Hence, Cu^2+^ ions have a critical contribution in improving chemotherapy based on DSF, and it has also been revealed that the appropriate amount of Cu^2+^ ion at the site of the tumor may have an important contribution to enhance DSF chemotherapeutic potential [23]. To elevate the level of Cu^2+^ ion at the site of the tumor and promote the therapeutic potential against the tumor, the presently provided approach works based on oral-administrated compounds (containing Cu^2+^ ion) that have usually been employed for treating Cu^2+^ ion deficiency in treatment centers [25]. However, the underlined treatment may result in the accumulation of Cu^2+^ ion (elevated level) in normal tissues, which causes heavy metal toxicity in those patients having a sufficient amount of copper [27,28,29,30]. Usually, the copper having high bioavailability is associated with proteins and other ligands as well. In each cell-free copper there exists less than one atom, and a surplus amount of free copper can cause poisoning at a high level [26,31]. Therefore, if we want to use chemotherapy based on the cuprous ion, it is essential to confine it to the cells of the tumor for avoiding the production of free copper ions in circulation. A very significant approach is required immediately to elevate the dose of copper ions in tumor cells and to remarkably attain the distinctive potential for the treatment of tumor and carcinoma. Recently, as a new drug release platform, light-responsive hydrogel has caused much attention. [32] Meng at al. designed a light-triggered in situ gelation to enable potent photodynamic-immunotherapy [33]. Notably, the drug release rate may be controlled by transforming the laser power density and the exposure time to realize the dissolution of the hydrogel. Furthermore, compared with some biocompatible and biodegradable inorganic nanostructures such as BP [34,35], the light-controlled reversible phase transition of the hydrogel can be used to deliver drug repeatedly and with minimal invasiveness.

In the current study, we designed a new Cu^2+^ release controller (CRC) system for cancer therapy. The hydrogels of agarose were used to wrap hierarchical microparticles (HMP) and CuCl_2_ and form the CRC system, as depicted in Scheme 1. Agarose is a safe and US-FDA-approved material [32], and HMPs were employed as the PTA (photothermal transducing agents) because of their high potency of transforming light into thermal energy [36,37]. When the system irradiation (at 808 nm) has been carried out for a few minutes, the HMPs transfer light energy into heat energy, and the hydrogel achieves reversible hydrolysis owing to overheating effects. Copper ions could be diffused into the microenvironment of a tumor from the hydrogel, which reacts with the DSF (pre-injected) and causes very harmful effects. The underlined controlled burst release may be useful to maintain the released drugs in the therapeutic window and effectively lowers the harmful effects caused by copper. 

## 2. Materials and Methods

Agarose, copper chloride (CuCl_2_), sodium hydroxide (NaOH), fluorescein diacetat (FDA), propidium iodide (PI), and cell counting kit-8 (CCK-8) were obtained from Sigma-Aldrich (USA). Phosphate Buffer Saline (PBS) was purchased from Thermo-Fisher (USA). All reagents used in this work were analytical reagents (A.R.) and used without any further purification.

### 2.1. Cell Culture

4T1 mouse breast cancer cell lines were obtained from the Cell Bank of the Chinese Academy of Sciences and incubated in Roswell Park Memorial Institute (RPMI)-1640 medium supplemented with 10% Fetal Calf Serum (FBS) in a humidified atmosphere at 37 °C with 5% CO_2_.

### 2.2. Isolation of Hierarchical Microstructures

The HMP was prepared by using the base extraction method. Briefly, 2 g hair was heated in a solution of 1 M NaOH wit slight stirring for 5 min. Then, the solution was dialyzed against PBS for 24 h. Next, the solution was centrifugated at 2000 revolutions per minute (RPM) for 6 min to remove large particles and 12,000 R9PM for 10 min to obtain the HMP. Finally, the HMP was washed with Deionized (DI) Water several times and dried for later use.

### 2.3. Synthesis of Different Hydrogel

To obtain pure hydrogel (Hy), 20 mg agarose was dispersed in 10 mL of deionized water in 50 °C. To obtain CRC, 20 mg agarose, 10 mg CuCl_2,_ and 3 ml HMP were dispersed in 10 mL of deionized solution in 50 °C. To obtain pure 0.8% Hy, 8 mg agarose was dispersed in 10 mL of deionized water in 50 °C.

### 2.4. Structural Characterizations of the Hydrogel

The morphology of nanoparticles was characterized using a transmission electron microscope (TEM; JEM-2010 ES500W, Tokyo, Japan) and field-emission scanning electron microscopy (Zeiss Merlin Compact). Zeta potential was detected by the dynamic light scattering (Nano-ZS ZEN3600). The Cu content was quantitatively analyzed by ICP-MS. UV−Vis−NIR spectra for hydrogel, HMPs, and CuCl_2_ were collected on a Cary 5000 UV−Vis−NIR spectrometer. 

### 2.5. Rheological Test

Rheology experiments were performed on an Anton Paar rheometer. Hydrogel samples of different temperatures were prepared and gently placed on the middle of a 15 mm diameter parallel plate with a proper gap. Dynamic oscillatory frequency sweep measurements were conducted at a 1% strain amplitude. To prevent the evaporation of water, a lid was prepared on the top.

### 2.6. Photothermal Conversion Efficiency

An 808 nm NIR laser (Changchun New Industries Tech.Co., Ltd., Changchun, China) with irradiation powers was used to stimulate the different concentrations (50, 100, and 200 ug/mL) of CRC in an aqueous medium. The photothermal images of the HMP-based suspensions during laser irradiation were recorded every 30 s using an infrared thermal imaging system. The NIR laser source was equipped with a 4 mm diameter laser module with an adjustable power. The photothermal conversion efficiency was calculated using the following equation:(1)$$\eta =\frac{hS\left(T_{max}-T_{surr}\right)-Q_{0}}{I\left(1-10^{-A_{\lambda }}\right)}$$
where *h* is the heat transfer coefficient, *S* is the surface of the container, *T_max_* and *T_surr_* are the equilibrium temperature and ambient temperature, respectively. *Q*_0_ is the heat associated with the light absorbance of the solvent, *A_λ_* is the absorbance of HMP at 808 nm, and *I* is the laser power density. According to the above equation, the η value of HMP was determined to be about 40.2%.

### 2.7. In Vitro Phototoxicity of CRC

The phototoxicity was measured by CCK-8 assay. 4T1 cells were seeded in 96-well plates at a density of 5 × 10^3^ cells per well and incubated for 24 h. Afterwards, 4T1 cells were incubated for 2 h with five different groups: (1) PBS + Laser (L), (2) Hy, (3), DSF + Hy, (4) Hy + L, and (5) DSF + Hy + L. Then, cells in group 1, 4, and 5 were irradiated with the 808 nm laser at a power density of 1 W/cm^2^ for 5 min. DSF concentration is 5 μg/mL. At the end of the incubation, 5 mg/mL CCK-8 PBS solution was added, and the plate was incubated for another 4 h. Finally, the absorbance values of the cells were determined by using a microplate reader (Emax Precision, San Jose, CA, USA) at 450 nm. The background absorbance of the well plate was measured and subtracted. The cytotoxicity was calculated by dividing the optical density (OD) values of treated groups (T) by the OD values of the control (C) (T/C × 100%).

To further visualize the cell phototoxicity of each group, 4T1 cells were incubated for 2 h with five different groups: (1) PBS + L, (2) Hy, (3) DSF + Hy, (4) Hy + L, and (5) DSF + Hy + L. Then, cells in group 1, 4, and 5 were irradiated with the 808 nm laser at a power density of 1 W/cm^2^ for 5 min. Then, all cells were washed with PBS, treated with FDA and PI according to the manufacturer’s protocol, and detected under a fluorescent microscope (IX81, Olympus, Japan).

### 2.8. In Vivo Toxicity

Healthy Balb/c mice were intravenous injection (i.v.) njected with PBS and DSF + CRC + L (10 mg/kg DSF, 10 mg/kg CRC; *n* = 5) or PBS. At 15th day post the injection, the major organs including heart, liver, spleen, lung, and kidney were harvested, fixed in 4% of formalin, embedded in paraffin, sectioned into 4 μm slices, stained with hematoxylin and eosin (H&E), and observed by an optical microscope (BX51, Olympus, Japan).

## 3. Results

### 3.1. Characterization of CRC NPs

First, the usable sample was prepared by diluting Cu^2+^ and HMP to a suitable concentration. Then, different kinds of hydrogels were prepared in a quick manner, as depicted in Figure 1A. It might be observed that all hydrogels prepared in a centrifuge tube did not move downward, which reveals the gelation of various solutions. HMP (rod-like or oval-shaped) might be observed with an average width (about 300 nm) and an average length (around 1 μm), as depicted in Figure 1B. Scanning electron microscope (SEM) images of the CRC revealed a complicated hydrogel pore size distribution, as indicated in Figure 1C. NIR-light irradiation of 1 W/cm^2^ 808 nm was employed for the study of the CRC temperature control characteristics (Figure 1D). First, the CRC exhibited light green colors, and with the persistent laser irradiations, CRC faded via degrees, indicating the degradation of CRC. Infrared thermal imaging also revealed that the CRC (irradiated via the laser) temperature was elevated, as depicted in Figure 1D. The rate of drug release in successive ON–OFF cycles (first four) revealed that CRC works as an operative optical switch for the release of drugs, as indicated in Figure 1E. Furthermore, agarose could be degraded post-treatment, emphasizing its ability for clinical uses. CRC rheology measurement was carried out with various concentrations of agarose, which showed that the storage modulus was decreased at an elevated level of temperature, as indicated in Figure 1F,G. As the temperature was elevated, the hydrogel storage modulus was reduced, which revealed an effective generation of the hydrogel. There were no considerable alterations in the zeta potential of CRC in the succeeding three days, which indicates the high stability of the underlined particles (Figure 1H) and is very important because various micro-platforms and medical materials have been associated with their instability, which would exert influence on clinical application.

### 3.2. Photo-Thermal of the CRC for PTT

It was identified that the chemical constituents of HMP are melanin and keratin [37]. Because of the melanin efficient photo-thermal conversion, HMP has a continuous and elevated photo-thermal conversion potential, due to which it is considered to be the ideal material with which to accomplish photo-thermal therapy (PTT). The HMP photo-thermal potency was evaluated via irradiating a centrifuge tube comprising different concentrations of aqueous HMP, such as 0.0, 10.0, 25.0, 50.0, and 100.0 μg/mL, along with an NIR laser (808 nm, 1 W/cm^2^) in parallel, whilst during irradiation, the impact of temperature was validated via taking the HMP solutions’ infrared thermal images, as shown in Figure 2A,B. The HMP photo-thermal heating effect for a persistent irradiation power depends on the concentration of HMP. Its increased concentration tends to an elevated heating effect, which reveals that the conversion of light energy to thermal energy via HMP is highly efficient. Additionally, in 5 min, the HMP solution (at 100 μg/mL) temperature was elevated from 30 (the starting temperature) to approximately 50 °C. This demonstrated that the laser irradiation produces hyperthermia and the elevated temperature is appropriate to defeat the tumorigenic cells via cleavage and denaturation of DNA, RNA, and intracellular protein. The HMP solution UV−Vis−NIR absorbance spectrum is depicted in Figure 2C, which shows a broad and intense peak of absorbance between 750 and 900 nm, without any other observable peaks. Based on these observations, the CRC hydrogel is an effective and appropriate photo-thermal material. Moreover, for 5 min, the HMP solution irradiation was carried out at a wavelength of 808 nm with 1 W/cm^2^, followed by turning off the laser to recover the starting temperature (i.e., 30 °C). This cyclic process was repeated four times (Figure 2E) to reveal that the temperature difference of the peak in each cycle was much less. The repetition of the underlined process also demonstrated the stability and reproducibility of the HMP photo-thermal performance. The HMP photo-thermal conversion potency (η) was determined from the obtained data, shown in Figure 2D,F. The calculated value of η was calculated to be 40.2%, and is expressively greater, in comparison with graphene quantum dots, Ti_3_C_2_ nanosheets, and Au nanorods values, i.e., 28.58%, 30.6%, and 21%, respectively.

### 3.3. In vitro Combination Therapy

Under different conditions, the fluorescent images of the 3T1 cells were evaluated and stained with FDA (fluorescein diacetate, green fluorescence) and PI (propidium iodide, red fluorescence) for live and dead cells, respectively, as depicted in Figure 3A. The CRC group fluorescence images were compared in the presence and absence of laser irradiation, and it indicated that 4T1 cell viability was effectively reduced by laser irradiations, as the elevation in temperature activates the hydrogel dissolution and release of Cu^2+^. The CRC groups and DSF were exposed to the laser irradiations, which indicated an elevated level of cell death because of CuET formation. The hydrogel (with DSF) by laser irradiations caused toxic effects on cells, as indicated in Figure 3B. After incubation of cells with hydrogel, the CRC + laser (L) group cells were dramatically affected, resulting in the cell death post 5 min of laser irradiation. The photo-thermal therapy in combination with DSF serves the maximum apoptotic potency against tumorigenic cells. Remarkably, the release of Cu^2+^ could be controlled, which is significant, because cells need a very low number of free copper ions, while an elevated level of free copper results in dramatic toxicity for healthy organs. Furthermore, we evaluate in vivo toxicity; sixteen days after the first injection of PBS or treatment with DSF + CRC + L, the mice were sacrificed and the liver, lungs, spleen, heart, and kidneys were examined for histological evidence of toxicity. No pathological changes or inflammation were observed in these organs (Figure 4), which showed the potential biosafety of this hydrogel combined with DSF treatment.

## 4. Conclusions

In summary, we have planned a relatively simple and significant approach to elevate photothermal therapy (PTT) efficiency of HMP- and DSF-mediated chemotherapy by developing a smart light-controlled drug-release CRC microstructure. This hydrogel drug delivery approach is composed of PTA HMPs, agarose, and DSF. In vitro experiments demonstrated that CRC could realize a manageable light-triggered Cu^2+^ release and hydrogel degradation. More significantly, the drug release rate can be specifically modified via internal (i.e., drug and agarose concentration) and external parameters (i.e., exposure duration and light intensity), which has a key role in clinical applications for the maintenance of an appropriate drug concentration in blood for the treatment of cancer. Taken together, the CRC-based drug delivery approach is a highly significant and candidate therapeutic approach against cancer and might decrease systemic toxicity of copper ion therapy.

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
