# Peer review of "Injectable Hydrogel for Cu2+ Controlled Release and Potent Tumor Therapy"

_life, 2021, doi:10.3390/life11050391_

Round 1
Reviewer 1 Report
The subject of the presented work is of interest to the journal and presents a new approach to oncological therapy with the drug disulfiram. The authors developed an interesting way to regulate the level of Cu2+ copper ions, which support the activity of the drug, but at the same time are harmful in excess. The work is well planned and done. In fact, I have no criticisms except for some minor suggestions:
- Figure 4B should be corrected - the present form of the drawing is not legible,
- there are typos in the text, e.g. lased irradiations (line 238),
- and the texts in the final part of the work, marked in yellow, should be corrected or deleted.
Author Response
We would like to thank the reviewer for pointing out the weakness in our manuscript. Typos and other problems in the text has been corrected in the current version according to your suggestions.

Reviewer 2 Report
Dear Editor in Chief,
Dear Authors,
I will start congratulating to the authors for the nice manuscript. I do not have any criticism or major concerns. the manuscript can be published but, if possible, I would suggest to the authors to indicate, wether suitable, to indicate is a simultaneous release of n antitumoral drug is possible!? This would be beneficial because in this way 2 symbiotic mechanism would act!
Best regards,
R2
Author Response
Thank you for your suggestions and questions. Our system can be used as a carrier for multiple drugs simultaneously.

Reviewer 3 Report
I have reviewed this manuscript entitled "Injectable Hydrogel for Cu2+ Controlled Release and Potent 2 Tumor Therapy’’. Disulfiram (DSF) is a crucial important drug for the treatment and management of alcohol dependency. Authors have designed an injectable hydrogel, as CRC (Cu2+ release controller) for effective tumor treatment. The hydrogels of agarose have been employed to wrap CuCl2 and hierarchical microparticles (HMP) for the generation of CRC system. Hence the CRC system appears as a novel approach for designing of Cu2+ delivery system against cancer and might reduce the systemic toxicity caused by copper ion therapy. The manuscript is written well and presented in a good manner, but still, it requires some changes like:
- Graphical abstract doesn’t represent the main idea of the manuscript.
- Please check the grammar, syntax of your revised manuscript before resubmission.
Author Response
Thank you for your comments and questions. We have made the modification according to your request.
